# Sugarcane Productivity Mapping through C-Band and L-Band SAR and Optical Satellite Imagery

**Ramses A. Molijn [1],\* , Lorenzo Iannini [1], Jansle Vieira Rocha [2] and Ramon F. Hanssen [1]**

[1]  Geoscience and Remote Sensing, Delft University of Technology, 2628 CN Delft, The Netherlands; l.iannini@tudelft.nl (L.I.); r.f.hanssen@tudelft.nl (R.F.H.)

[2]  Faculdade de Engenharia Agrícola (FEAGRI), Unicamp, Campinas 13083-875, Brazil; jansle@unicamp.br

\*  Correspondence: r.a.molijn@tudelft.nl

**Abstract:** Space-based remote sensing imagery can provide a valuable and cost-effective set of observations for mapping crop-productivity differences. The effectiveness of such signals is dependent on several conditions that are related to crop and sensor characteristics. In this paper, we present the dynamic behavior of signals from five Synthetic Aperture Radar (SAR) sensors and optical sensors with growing sugarcane, focusing on saturation effects and the influence of precipitation events. In addition, we analyzed the level of agreement within and between these spaceborne datasets over space and time. As a result, we produced a list of conditions during which the acquisition of satellite imagery is most effective for sugarcane productivity monitoring. For this, we analyzed remote sensing data from two C-band SAR (Sentinel-1 and Radarsat-2), one L-band SAR (ALOS-2), and two optical sensors (Landsat-8 and WorldView-2), in conjunction with detailed ground-reference data acquired over several sugarcane fields in the state of São Paulo, Brazil. We conclude that satellite imagery from L-band SAR and optical sensors is preferred for monitoring sugarcane biomass growth in time and space. Additionally, C-band SAR imagery offers the potential for mapping spatial variations during specific time windows and may be further exploited for its precipitation sensitivity.

**Keywords:** sugarcane growth monitoring; SAR and optical remote sensing; precipitation effects; saturation effects

## 1. Introduction

Sugarcane is the number one globally cultivated crop in terms of production quantity, more than the second and third crops, maize and rice, combined. Brazil is the largest sugarcane producer, amounting to almost 40% of total global production [1], while São Paulo state hosts more than 60% of Brazil's sugarcane acreage [2,3]. Over the last 15 years, this acreage more than doubled [4]. The main products are sugar and bioethanol; the latter allows Brazil to reduce the national automotive's gasoline consumption by more than half [2]. This illustrates the importance of sugarcane production and research in São Paulo, which hosts the area of study.

Sugarcane is a semiperennial crop; after each growth cycle, typically lasting for 12 to 18 months for Brazilian plantations [5], new ratoons emerge from the same root system. Since the yield decreases over time, the plants and the root system are generally removed after five to seven years, and new shoots are planted. Indicatively, the growth cycle follows phenological phases as depicted in Figure 1. During the last phase, the sucrose content in the stem accumulates and senescence of the leaves occurs [5–7].

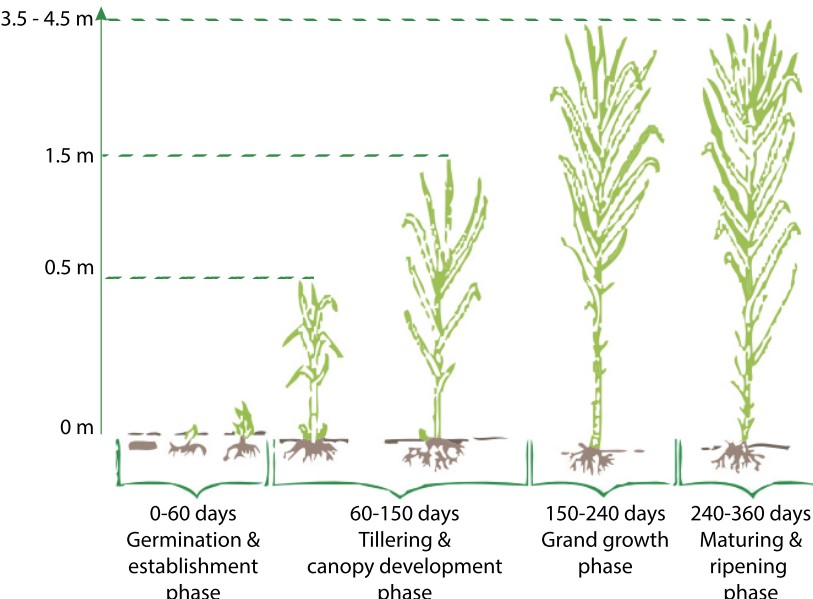

**Figure 1.** Phenological phases of sugarcane and their corresponding time frames, as well as indicative heights and plant geometries. Adapted and modified from NaanDanJain [8].

The sensitivity of remote observations to crop conditions was largely investigated for crop-monitoring uses [9–11] as well as to improve the understanding of the plant–microwave interaction [6,12]. Review works by McNairn and Brisco, Steele-Dunne et al. [13,14], and a study by Moran et al. [12] give a general introduction on the factors that affect Synthetic Aperture Radar (SAR) signals over the course of crop growth. The signal backscatter is mainly influenced by three mechanisms: the interaction of the microwaves directly with the vegetation, directly with the soil, and the interaction between vegetation and the underlying soil. Regarding the vegetation contribution, the backscatter coefficient is governed by its dielectric properties and its geometrical aspects. Its dielectric properties are mainly dependent on the vegetation water content and water droplets on the plant after precipitation events. The contribution from the soil is similarly governed by its geometry (roughness and slope) and by its moisture. The backscatter coefficient is further dependent on the configuration of the sensor, including viewing geometry, wavelength, and polarization. The soil backscatter contribution for C-band SAR is small from the leaf-development stage onward [12], while it is more significant for L-band SAR signals with standing vegetation [14]. In addition, the impact of the incidence angle on backscatter was found to be minimal compared to the effects from soil and vegetation conditions for C-band SAR [12], but more apparent for L-band SAR [6].

The monitoring capabilities of sugarcane growth with optical signals was described by several studies mainly focusing on biophysical-parameter extraction [3,9,15–17] and yield estimation [10,16,18,19]. Varying rates of success were reported, and the main limiting factors were attributed to the scarce availability of optical images and the complex relation between observed signals and the estimated parameter or feature. By integrating agrometeorological data with vegetation indices, better results were achieved, though still not operationally applicable due to challenges in the agrometeorological modelling [10]. Another solution was found in the use of optical data acquired by unmanned aerial vehicles at the time of interest [9]. This study also showed that spatial patterns in height estimations from optical signals were in agreement with observed heterogeneities in the ground-reference data. Furthermore, a short study on the value of C-band SAR (Radarsat-2 and Sentinel-1) and optical imagery (Landsat-8), acquired over sugarcane fields [20], found that SAR imagery contains fewer spatial features that are statistically significant and persistent in time than optical imagery.

Far fewer studies were published on the capabilities of SAR signals for the mapping and monitoring of sugarcane than were published for other crops like wheat, barley, and maize. Commonly, the works on sugarcane cover the use of SAR signals for retrieving sugarcane height and harvest events. Two of the foremost publications [6,21] show that X-band signals from TerraSAR-X and C-band signals from ASAR saturate at a lower sugarcane height than L-band signals from ALOS and NDVI signals from SPOT sensors. It was observed that X-band signals experience saturation between 0.5 and 1 m, and C-band signals after 1 m. L-band signal backscatter increased significantly until 1.5 m, moderately from 1.5 to 2 m, and marginally from 3 m onward. In addition, a decrease in NDVI and C-band radar backscatter close to harvest was demonstrated, which was linked to the severe drying of the sugarcane. This drop was found to be on the same order for harvested fields where no (assumed) drying occurred before harvest, leading to ambiguity on the detection of harvest events. For L-band signals, a clear relationship was observed between HH backscatter and NDVI over the course of sugarcane growth. Here, the authors relate their simultaneous declines to water stress of growing maturating sugarcane plants.

Other works, led by Unicamp in Brazil, focused on the behavior of ALOS signals with changing sugarcane conditions [11,22,23]. It was found that signals can be used to discriminate between the first phases of sugarcane growth until the grand growth phase commences, after which the signals saturate. In addition, it was concluded that the backscatter acquired in HH increased significantly when rows perpendicular to the look direction were viewed (in the order of 1.2 dB), similar to what was observed in Baghdadi et al. [6], but only for the first eight months (until the last phenological phase). This phenomenon is also named 'flashing field' in the literature. No significant effect of the row direction was observed for HV signals. Emphasis was placed on the need of including rain events when interpreting the signals because of the significant signal increase, although this was not quantified.

In light of these literature findings, we identified a scarcity in published research on the extraction of spatial patterns from remote sensing imagery acquired over crops, especially sugarcane, their spatial consistency over time, and the effects from precipitation and varying beam-acquisition modes. The temporal development of spatial patterns in relation to the ground-measured biophysical parameters of sugarcane is also underexposed. Accordingly, in this manuscript, we first demonstrate the effects of sugarcane conditions on optical and SAR signals, with specific focus on biomass growth and precipitation events. Subsequently, we extracted the spatial patterns that are most consistent in time and compared these patterns between the different sensors in order to discover their agreements and differences. Finally, we related these patterns to intrafield sugarcane-biomass variations. The term consistency is commonly used in this document as an abbreviated denotation for the temporal consistency of spatial features in remote sensing imagery, which is further mathematically defined in Section 3.2.

Since the mentioned literature shares the finding that cross-polarization signals are generally more effective in mapping and monitoring sugarcane than copolarization signals, emphasis is put on the analysis of HV signals in this paper; HH signals are addressed, but not with the same level of detail.

## 2. Study Area and Data

### 2.1. Fields of Interest

The four sugarcane fields of interest are owned by one of the largest sugarcane producers and energy companies in Brazil. They are located 20 km from Piracicaba in São Paulo state. All fields are within 15 km from each other (see Figure 2), connected by interfield roads of two to ten meters wide. The agriculture in this region is dominated by sugarcane plantations and is relatively flat. The fields vary in size, ratoon cycle, and time of growth and harvest (see Table 1), but have the same soil type.

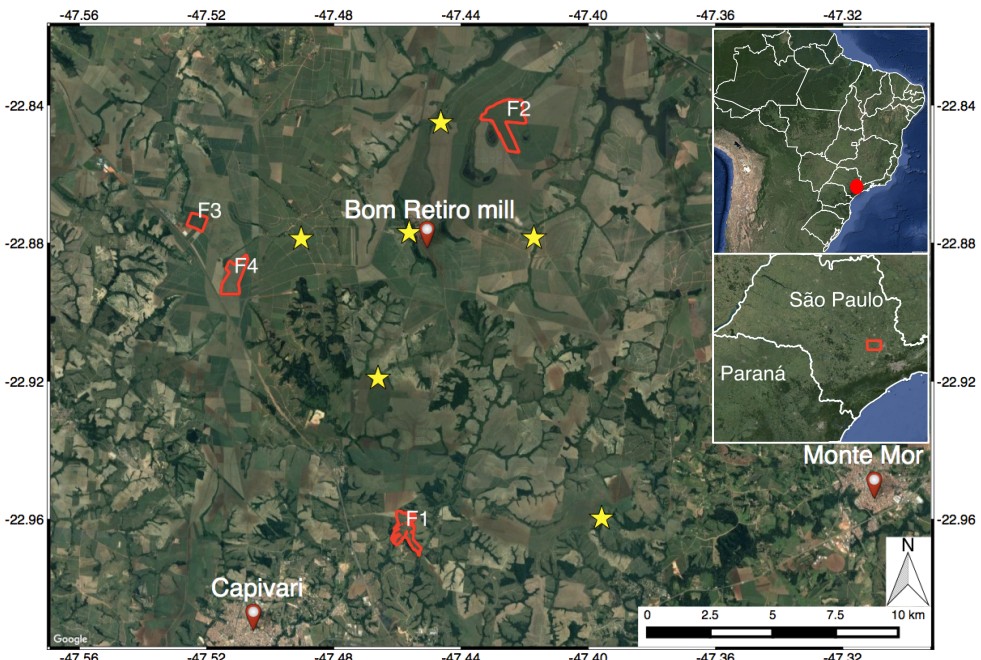

**Figure 2.** Location of the four fields of interest, indicated by red polygons, and weather stations, indicated by yellow stars.

**Table 1.** Characteristics of the four sugarcane fields; growth and harvest dates are not applicable to the entire field for F1 and F2.

| Field Name | Ratoon | Area (ha) | Start of Growth | Harvest | Center Coordinate (Latitude, Longitude) |
|---|---|---|---|---|---|
| F1 | 1st cycle | 58 | 30/10/2014 | 07/10/2015 | $-22.9607°$, $-47.4578°$ |
| F2 | 1st cycle | 115 | 14/10/2014 | 07/12/2015 | $-22.8558°$, $-47.4228°$ |
| F3 | 2nd cycle | 25 | 15/08/2014 | 26/07/2015 | $-22.8738°$, $-47.5230°$ |
| F4 | 9th cycle | 59 | 01/08/2014 | 21/07/2015 | $-22.8895°$, $-47.5108°$ |

### 2.2. Ground-Reference Data

From October 2014 to October 2015, an extensive set of ground-reference measurements were taken, amounting to a total of more than 3500 individual measurements on LAI, biometrics (including stalk height and thickness, leaf length, and cane spatial density), and biomass parameters (including cane water content and cane mass density). In Molijn et al. [24] we present in detail the methodologies for acquiring the measurements, the measured parameters, and the applied modelling and uncertainty analyses. The dataset itself was published for public use [25]. To summarize, two types of measurement locations were selected, spread over four fields (see Table 1 and Figure 3). The first type of locations consists of 15 Elementary Sampling Units (ESUs) that were selected based on the intrafield variability seen in Landsat-8 (LS8) NDVI images of previous years. The second type consists of six ESUs for biomass (ESUBs). Each location covers approximately an area of 20 by 20 m and was, on average, revisited 15 times during the growth period (see also Section 2.3). At the ESUBs, LAI and biometrics, consisting of sugarcane thickness, height, and cane spatial density, were measured, as well as dry and wet biomass. At the ESUs, only biometric measurements were taken and used as input parameters in a biomass-estimation equation, which was calibrated and validated by the measured biomass samples. This methodology allows for estimating, in a nondestructive manner, the sugarcane biomass in each ESU. Since only biometrics and LAI were measured at these locations, ground conditions were not affected and could be used for the analysis of the remote

sensing signals. The goodness of fit, expressed as the coefficient of determination, between estimated and measured biomass was 0.90, which gave us sufficient confidence in the validity of the method for estimating biomass at the required ESUs for this study. Biomass samples were also used for retrieving cane mass density and leaf mass per cane.

It should be noted that, for Fields F2, F3, and F4, the start of growth dates apply for the entirety of each field. For Field F2, sugarcane harvesting lasted for several weeks due to delays. Consequently, we analyzed this field until its first harvest event. For Field F1, certain parts of the field were uniformly grown over the course of the remote sensing observations. The parts that hosted ESUs and ESUBs were extracted and individually incorporated in the analyses based on their start of growth dates.

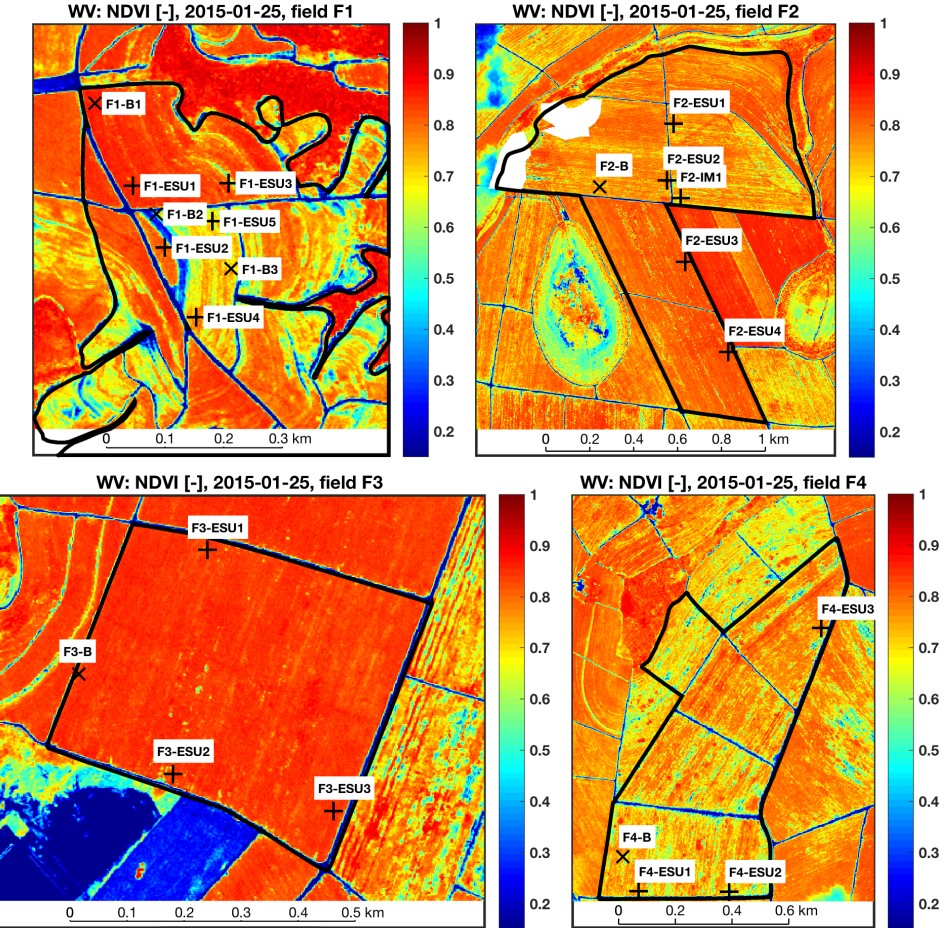

**Figure 3.** Delineation of the four studied sugarcane fields. Background map shows NDVI from WorldView-2 on 25 January 2015. Plus signs (+) and crosses (×) show, respectively, the location of the elementary sampling units (ESUs) and elementary sampling units for biomass (ESUBs). White patches in Field F2 are masked clouds. North direction is up.

In this manuscript, sugarcane biomass is defined as cane biomass, which is the the biomass of a single stalk (i.e., stem) together with the biomass of its leafs. The product of cane biomass with cane spatial density gives the biomass per unit area, commonly expressed as tons cane per hectare (TCH) [26–29]. The TCH estimates of all four fields were used for backscatter analysis (Section 3.1). For subsequent analyses, only the TCH estimates of Field F2 were used due to three reasons. First, the time series of the satellite imagery covers the entire growth period of this field as opposed to Fields F3 and F4. Second, it is

the only field with four ESUs (see Figure 3), and in this field the largest intrafield TCH differences were measured (see the left part of Figure 4). Here, the figure's uncertainty bars represent one standard deviation on either side of the mean profile based on the computed uncertainties associated with the measurement error and local variability; see Molijn et al. [24] for more detailed explanations. Third, on 19 December 2014 (66 days after start of growth, cane height up to one meter) in Field F2, 24 intensive measurements were taken for an indication of intrafield biomass variability. In total, the locations were spread over 400 m with spaces of 5, 10, or 20 m north of point F2-IM1 (see Figure 3). The resulting biomass estimates along this profile (see the right part of Figure 4) showed gaps and peaks up to 50% higher than the field average TCH (approximately 30 tons per hectare) and no evident spatial trend. These variations are a direct result of variabilities in cane spatial density.

The development of a selection of the measured parameters during the ground campaign (see Figure 5) shows a deflection point for leaf length and stalk thickness during the canopy-development phase and a continuous increase of stalk height until harvest. The ratio of leaf biomass over cane biomass shows that leaf biomass is dominant during, approximately, the first 50 days (i.e., establishment phase), after which the stalk biomass dominates and the leaf biomass contribute rapidly declines. The main reason is the decline in leaf wet content and increase in stalk mass density combined with stalk growth [24].

Finally, daily cumulative precipitation measurements were collected by seven weather stations. At least one weather station was within 10 km distance of each field. In terms of detected rain, the stations were in 96% agreement with each other, based on which we made the assumption that all fields were subject to the same condition of being precipitated or not precipitated.

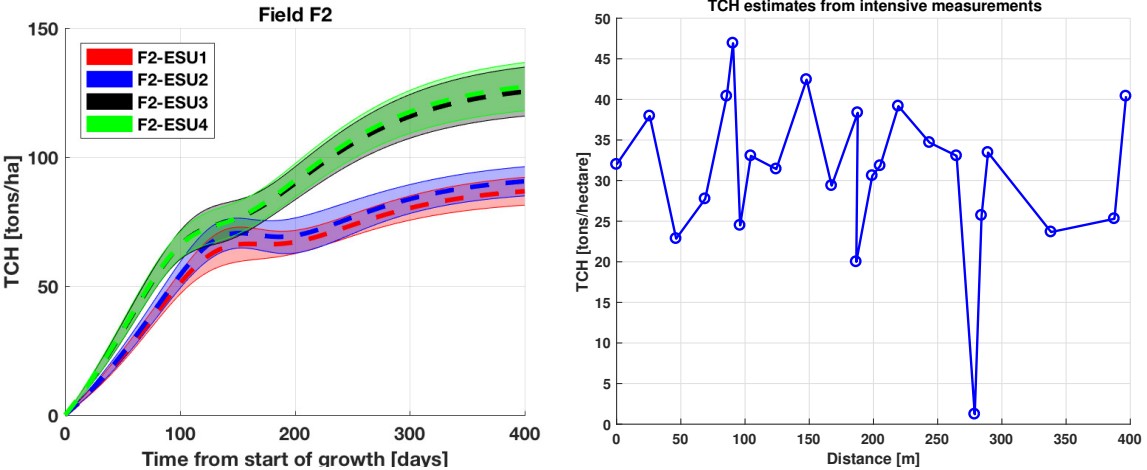

**Figure 4.** Estimated tons cane per hectare (TCH) over growth time for each elementary sampling unit (ESU) of Field F2, along with the associated uncertainties (**left**). F2-ESU1 and F2-ESU2 nearly overlap; similarly for F2-ESU3 and F2-ESU4. Profile of biomass estimates 400 m along from the intensive measurement at Field F2 (**right**).

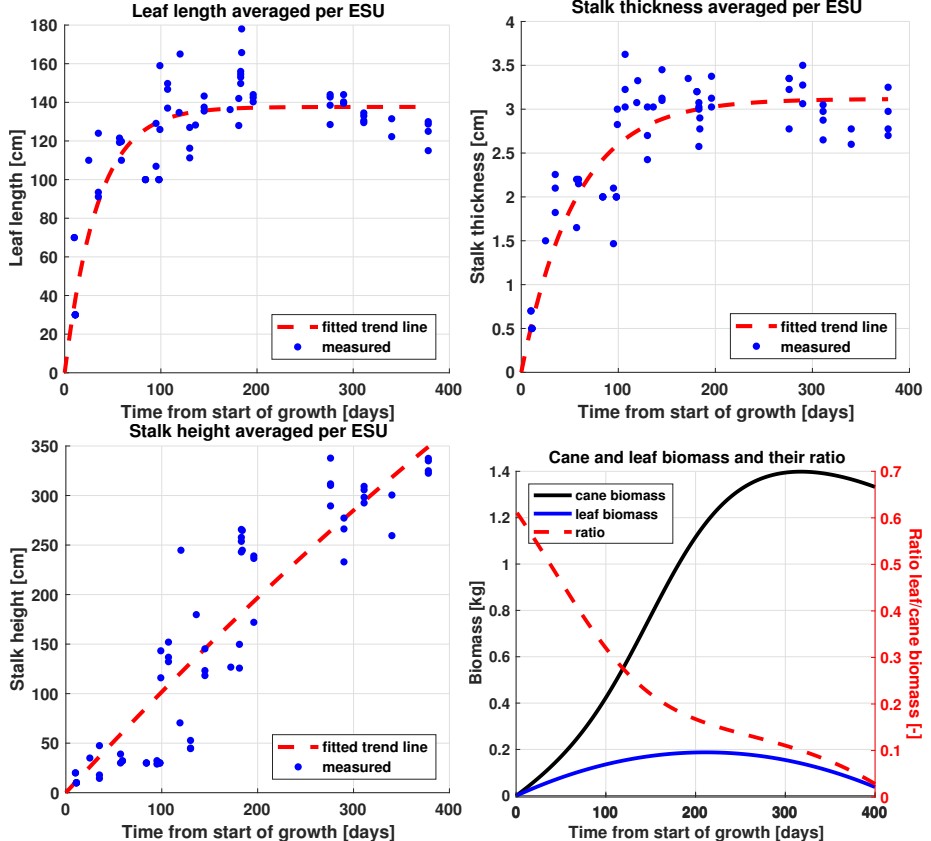

**Figure 5.** Temporal developments of measured leaf length, stalk thickness, and stalk height, averaged per ESU and accompanied by fitted trend lines. The cane and leaf biomass (in kg) per cane and their ratio over time are based on the trend lines of individual stalk- and leaf-biomass measurements.

## 2.3. Remote Sensing Data

### 2.3.1. Acquisitions and Preprocessing

Sensor characteristics vary in terms of wavelengths, polarizations, resolutions, and beam angles (see Table 2). Radarsat-2 (RS2) and ALOS-2 data were multilooked to a 30 by 30 m resolution. Sentinel-1 (S1) images were resampled to the same 30 by 30 m resolution grid. All radar images were radiometrically calibrated, geometrically corrected, terrain-corrected, and thermal-noise-corrected. SAR images were labelled as precipitation (or rain)-affected based on the time of acquisition and daily cumulative precipitation measurements from the nearest weather station. No precipitation threshold was applied.

The acquisition schemes of ground-reference and remote sensing data (Figure 6) show that the entire growth cycles are covered by ground and space measurements, except for SAR data during the first three months of Fields F3 and F4. Of the 29 LS8 acquisitions available, 11 had clear-sky observations for all fields. For all sensors, the number of acquisitions were counted between 1 July 2014 and 31 December 2015, i.e., just before the first start of growth and after the last harvest event of the fields. WorldView imagery consists of two WorldView-2 (WV2) images and one WorldView-3 (WV3) image. The WV2 images were acquired on 25 January and 11 August 2015. The WV3 image was acquired after the harvest events on 21 March 2016, and was used for additional visual inspection of the fields.

**Table 2.** Characteristics of available remote sensing data. [1] RS2 = Radarsat-2, S1 = Sentinel-1, LS8 = Landsat-8, WV = WorldView; [2] FQ: Fine Quad, W: Wide, S: Standard, EW: Extra Wide mode, SM: Stripmap, ASC: ascending pass, DSC: descending pass; [3] incidence angles; [4] UTC and approximate times, local time -2 or -3 hours; [5] regridded by ESA to medium resolution GRD product of 93 m × 87 m resolution; [6] five images with HV + HH, and three images with only HH; [7] approximate SLC product resolutions at field in meters; [†] rain label based on the daily cumulative previous to the day of acquisition; [*] rain label based on the daily cumulative on the day of acquisition. Reported resolutions and angles are mainly based on sensor and product descriptions [30–36].

| Sensor [1] | Instrument | Mode [2] | Polarizations/ Bands Used | Nominal Resolution (Slant Range × Azimuth) [7] | Angle Fields [3] (°) | Acquisition Frequency/Time [4] | Images |
|---|---|---|---|---|---|---|---|
| RS2 | C-band (5.5 cm) | FQ16W (ASC) | HH + HV + VH + VV | 5.2 × 7.6 | 35.7–36.3 | 24 days/21:45 [*] | 14 |
| | | FQ21W (DSC) | | | 40.9–41.4 | 24 days/08:32 [†] | 8 |
| | | S5 (ASC & DSC) | | | 40.9–41.4 | 24 days /21:45 [*] & 08:32 [†] | 2 |
| | | S6 (ASC) | HH + HV | 13.5 × 7.7 | 41.5–41.9 | 24 days/21:49 [*] | 5 |
| | | S7 (ASC) | | | 48.4–48.9 | 24 days/20:55 [*] | 9 |
| S1 | C-band (5.5 cm) | EW (DSC) | HH + HV | 11.5 × 43 [5] | 36.0–36.6 | 12 days/08:38 [†] | 33 |
| ALOS-2 | L-band (24 cm) | SM (ASC & DSC) | HH + HV or HH | 6.0 × 4.3 | 31.3–42.8 | 14 days /03:49 [†] & 14:43 [†] | 8 [6] |
| LS8 | Optical | Reflectances | RGB + NIR | 30 × 30 | NA | 16 days/10:10 | 29 |
| WV | Optical | Reflectances | RGB + NIR1 | <2 × <2 | NA | NA | 3 |

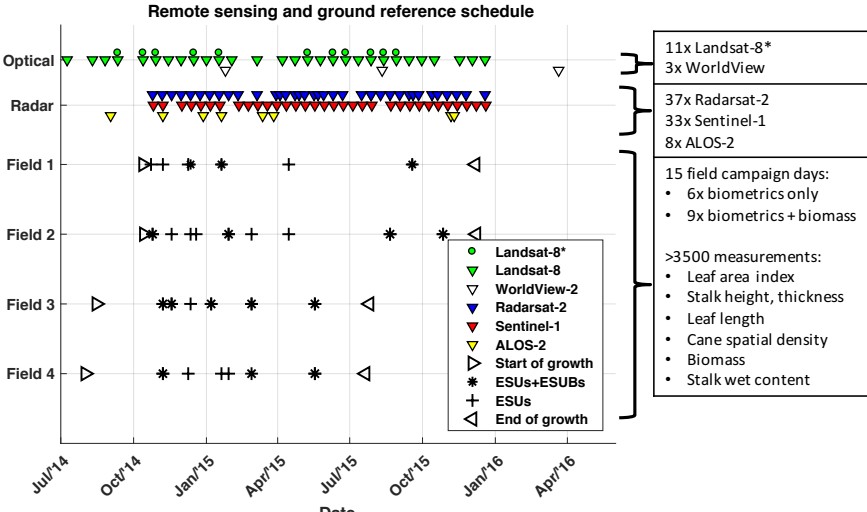

**Figure 6.** Temporal distribution of remote sensing images and ground-reference measurement dates. (Right box) summary of acquisitions, including number of image acquisitions per sensor, counted over growth duration. Landsat-8* symbols, LS8 acquisitions without cloud obstruction for all fields.

The remote sensing images of Field F2 taken by the different sensors, acquired approximately at the same date (see Figure 7), illustrate the spatial variabilities of the signals per sensor and their differences between sensors. Apart from the difference in backscatter magnitude between the C-band and L-band, the images were typically noisy and show no clear pattern agreement between the sensors. The most visually dominant field features included intrafield roads (low NDVI) versus the sugarcane plants (high NDVI). Outside the field, the most important feature was the water-drainage channel along the upper field border, as visible in the ALOS-2 image. This strip naturally contains more water, especially along the upper-right border due to elevation differences (see Section 2.3.3).

As described in Section 1, one of the factors causing variability in SAR backscatter intensities over vegetation is a changing incidence angle. This applies to the RS2 images as well due to the various acquisition modes. Even though the effect of incidence angle was reported to be minimal compared to the effects from varying soil and vegetation structure conditions, we applied an incidence-angle correction on RS2 images in order to more accurately compare the signals from RS2 and S1. Overall, considering the growth characteristics of sugarcane, we assumed high backscatter attenuation through the canopy based on the described literature and the high volumetric density of sugarcane. Consequently, we applied the following incidence-angle correction, based on the cosine dependency as proposed by Attema and Ullaby [37]:

$$\sigma^0_{RS2*} = \sigma^0_{RS2} \frac{\cos \theta_{S1}}{\cos \theta_{RS2}}$$

where $\sigma^0_{RS2*}$ is the beam-adjusted RS2 backscatter coefficient, $\sigma^0_{RS2}$ is the original RS2 backscatter coefficient, $\theta_{RS2}$ is the RS2 incidence angle over the fields (which differs per mode), and $\theta_{S1}$ is the S1 incidence angle over the fields (taken as 36.3°). Notice that this angle correction is equivalent to using the Gamma nought ($\gamma_0$)-normalized backscatter. As an illustrative example of the effects of this equation, we applied the correction on one S1 EW image (swath width 400 km) covering more than 40,000 sugarcane fields that were delineated by the Canasat project [4,38]. Before correction, the median signal backscatter ranged between −14.3 dB at a 29° and −16.1 dB at 47° incidence angle (see the left part of Figure 8). After correction, the spread of the median signal backscatter ranged from −14.6 dB at a 29° to −15.3 dB at 47° incidence angle; the median at 36.3° before and after correction remained −14.9 dB (see the right part of Figure 8).

Since the RS2 signal radiometric stability was smaller than 1 dB [30], we accepted this method for reducing the spread.

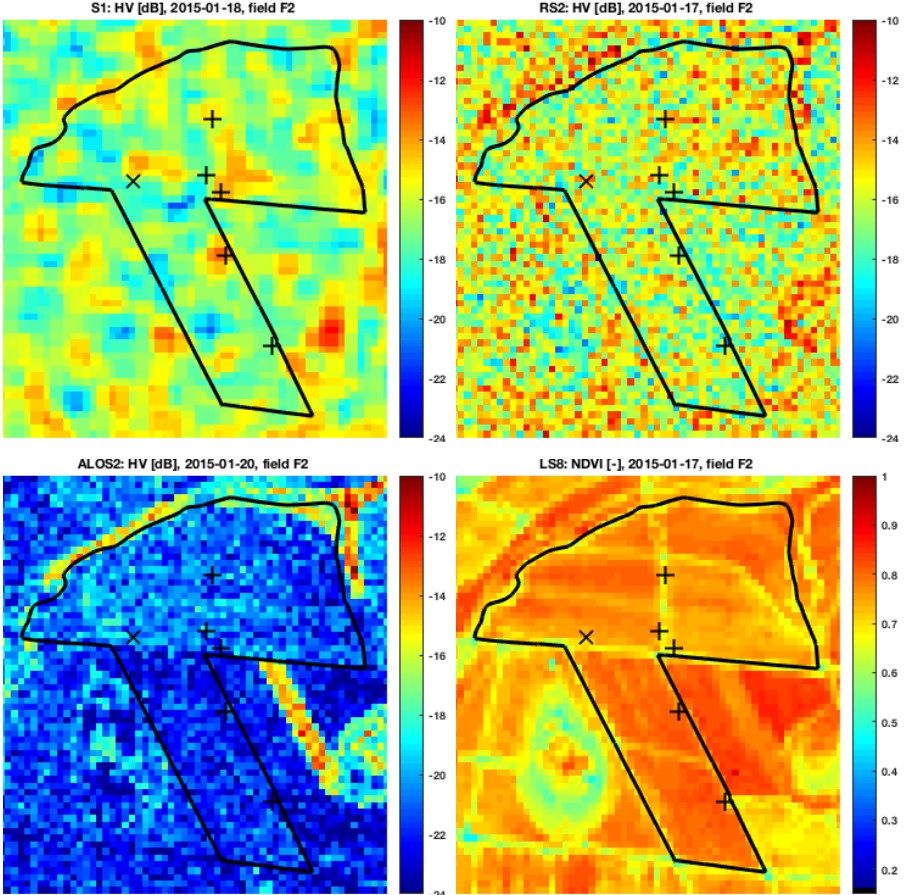

**Figure 7.** Sample images from available sensors, spatially averaged to 30 by 30 m pixel spacing, at approximately the same date for Field F2, around 100 days after start of growth, with a sugarcane height of 1 m. Plus signs (+) and crosses (×), location of ESUs and ESUs for biomass (ESUBs), respectively. North direction is up and field spans approximately 1.4 km northward by 1.7 km eastward.

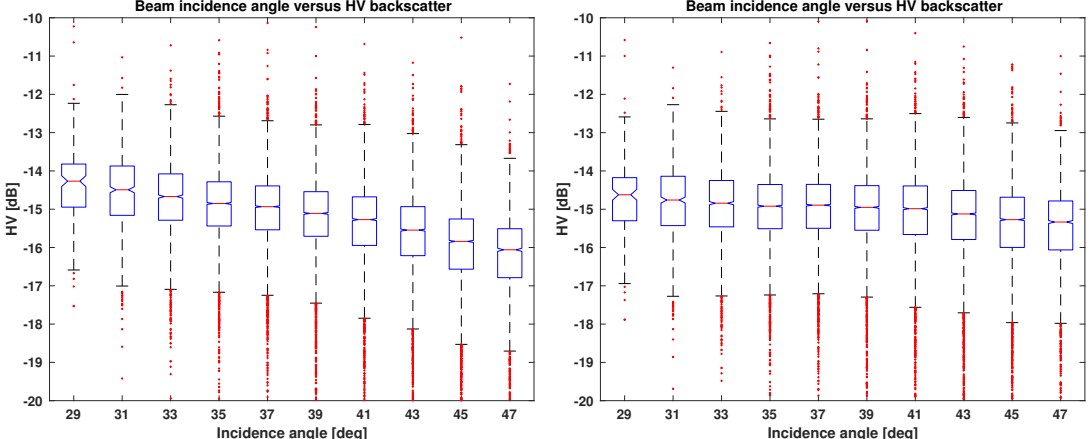

**Figure 8.** S1 EW HV backscatter versus incidence angles, before incidence-angle correction (**left**) and after incidence-angle correction (**right**).

2.3.2. Signal-Noise Budgets and Accuracies

For computation of the temporal consistency of spatial patterns from remote sensing imagery (as is further explained in Section 3.2), sensor-specific signal-noise budgets are quantified. For SAR sensors, we considered radiometric stability ($\varsigma_{RAD}$), maximum Noise Equivalent Sigma Zero ($\varsigma_{NESZ}$) and speckle noise ($\varsigma_{SPECKLE}$) as contributions to the total noise budgets. The first noise, $\varsigma_{RAD}$, acted as an multiplicative factor, whereas the second, $\varsigma_{NESZ}$, had an additive nature. Both were taken from nominal sensor and product specifications [30,32,33,39] (see Table 3). $\varsigma_{SPECKLE}$ is dependent on the number of looks and, hence, on the spatial averaging window. For intra- and intersensor imagery-consistency analysis, we resampled the signals of all sensors over a spatial window of 90 by 90 m (the approximate S1-EW GRD product resolution). Based on the original SLC ground resolutions, the resulting numbers of looks are 11 for S1-EW, 128 for RS2-FQ, 55 for RS2-S, and 186 for ALOS-2. For $\varsigma_{SPECKLE}$, it was assumed that the scattering medium in the averaging window was homogeneous. Total noise $\varsigma_{TOTAL}$ is quantified for three reference backscatter values that are indicative for the backscatter range found at the sugarcane fields (see Table 3).

As an explanatory example, the noise contributions of S1-EW were specified as follows. $\varsigma_{RAD}$ (expressed as $3\sigma$) is a constant equivalent to 0.5 dB, $\varsigma_{NESZ}$ is equivalent to $-22$ dB and needs to be treated as relative to the signal [40], contributing 0.1, 0.3, and 1.4 dB to the three reference signals of $-10$, $-15$, and $-20$ dB, respectively. $\varsigma_{SPECKLE}$ is dependent on the number of looks (eleven in this case), but not on the reference signal, and is equivalent to 1.4 dB. As a result, $\varsigma_{TOTAL}$ is approximately 1.7 dB for a backscatter coefficient of $-15$ dB, whereby $\varsigma_{SPECKLE}$ is the major contributor to the noise for high-valued signals and $\varsigma_{NESZ}$ becomes increasingly important for the low-valued signals.

**Table 3.** Signal-noise budget associated with data from different sensors and modes for 90 by 90 m spatially resampled SAR signals, with total noise computed for three backscatter intensity scenarios. Radiometric stability and NESZ values were extracted from sensor and product descriptions [30,32,33,39].

| Parameter | S1 | RS2-FQ | RS2-S | ALOS-2 |
|---|---|---|---|---|
| $\varsigma_{RAD}(3\sigma)$ | 0.5 dB | 1 dB | 1 dB | 1.2 dB |
| $\varsigma_{NESZ}$ | $-22$ dB | $-35$ dB | $-29$ dB | $-28$ dB |
| $\varsigma_{SPECKLE}$ | 1.4 dB | 0.4 dB | 0.6 dB | 0.3 dB |
| $\varsigma_{TOTAL}@-10$ dB | 1.5 dB | 0.5 dB | 0.7 dB | 0.5 dB |
| $\varsigma_{TOTAL}@-15$ dB | 1.7 dB | 0.5 dB | 0.7 dB | 0.5 dB |
| $\varsigma_{TOTAL}@-20$ dB | 2.7 dB | 0.5 dB | 0.8 dB | 0.5 dB |

With regard to the noise associated with NDVI from LS8, Akdim et al. [41] reported NDVI root mean square error (RMSE) values of 0.047 and 0.033 through comparing simulated NDVI values of LS8 with SPOT-4 images and LS8 with RapidEye images, respectively. To verify and improve these values, we assessed the temporal NDVI variation over stable vegetation, for which nearby native forest areas were selected, and computed the RMSE based on consecutive images with a maximum time difference of 32 days (i.e., maximum two revisits apart). We assumed that the differences of surface reflectances caused by the vegetation itself within this time frame were insignificant with respect to sensor noise. We also acknowledge that (illumination and sensor) geometry and topographic conditions affect NDVI variations [42,43] and contribute to this noise. Computation resulted in an additional RMSE of 0.027 and, through averaging the reported RMSEs (i.e., by taking the square root of the mean of the square of the RMSEs), we concluded on a total NDVI noise budget of 0.037.

Geolocation accuracies, defined as the RMSE estimated from point target monitoring, were smaller than 6 m for the relevant beam modes of S1, RS2, and ALOS-2 [44–47]. The geolocation products of the optical sensors are commonly specified using circular error at the 90th percentile (CE90), which, for LS8 is less than 20 m [48,49], and for the WorldView sensors this is less than 5 m [36,50,51].

### 2.3.3. Elevation

The widely used Shuttle Radar Topography Mission (SRTM) data product provided by NASA was used for the elevation information of the sugarcane fields. Acquired in 2000, the DEM is offered as a 30 m resolution product. Since it was derived based on C-band and X-band interferometric SAR (InSAR) techniques, several studies reported that errors in SRTM increase with increasing vegetation cover, especially for forests, due to the limited penetration of the X-band and C-band signals to the ground [52,53]. The methodologies these and similar works propose for correcting overestimations are based on vegetation masks and coarse-resolution-estimated vegetation heights. Hence, for this study we assumed these elevations could be used for interpreting relative in-field ground-elevation differences at our time and area of interest. The variations in elevation of Field F2 (Figure 9) show that adjacent to the boundaries of the upper part of the field, especially at the lowest-elevated places with the highest slopes, native vegetation strips are situated, serving as drainage systems for excess water discharge and water preservation.

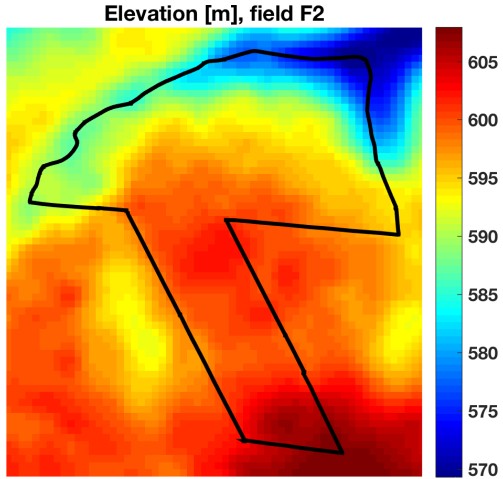

**Figure 9.** Elevation of Field F2. North direction is up and field spans approximately 1.4 km northward by 1.7 km eastward.

## 3. Methods

### 3.1. Backscatter Analysis

In order to investigate the effect of increasing biomass on remote sensing signals, the average of the modeled ESU biomass values was taken, expressed in TCH, per sugarcane field at each remote sensing acquisition. It is assumed that these averages are representative for the biomass conditions of the entirety of the corresponding field. Subsequently, the average of the remote sensing signals was taken per sugarcane field and plotted against TCH values. For each TCH bin, the average and standard deviation were taken for illustrating development and spread over time. Together with the TCH axis, the time of growth and stalk-height axes is visualized as well, though only for approximate and indicative purposes since these are not strictly linearly related to the TCH values. We defined the saturation point of the remote sensing signal as the first point in time when it does not increase with increasing biomass. For visualizing the effects of rain on SAR backscatter, box plots were produced based on SAR signals after saturation points.

*3.2. Intrasensor Imagery-Consistency Analysis*

For each sensor, the temporal coherence of the spatial features in remote sensing imagery was examined. For this, we introduce a spatiotemporal signal-consistency measure, based on which time windows with maximum consistencies are deduced. Remote sensing images were first spatially averaged over a 90 by 90 m window (approximately the S1 GRD product resolution, see Section 2.3). Subsequently, in order to extract pattern dynamics, we applied *z-score* normalization ($Z_{i,p}$) for each pixel ($p$) per image ($i$), by subtracting the field average of the signals ($\mu_i$) from the pixel value ($X_{i,p}$) and normalizing by the field standard deviation of the signals ($\sigma_i$):

$$Z_{i,p} = \frac{X_{i,p} - \mu_i}{\sigma_i} \tag{1}$$

For the temporal consistency of the images, we compared all $N_i$ available images from the start of growth until harvest with each other, i.e., for $\frac{N_i(N_i-1)}{2}$ possible combinations. Consistency metric $C$ is based on the ratio between log-likelihoods:

$$C = 1 - \frac{\log L_0}{\log L_1} \tag{2}$$

The first element, $\log L_0$, represents the log-likelihood of the null hypothesis that the two images belong to the same spatial distribution, and hence share the same spatial pattern, and is expressed by:

$$\log L_0 = -\frac{1}{N_p} \sum_{p=1}^{N_p} \frac{\left(Z_{i,p} - Z_{j,p}\right)^2}{\sigma_{Z_{i,p}}^2 + \sigma_{Z_{j,p}}^2} \tag{3}$$

The second element, $\log L_1$, represents the log-likelihood of the alternative hypothesis that the two images do not share the same spatial pattern, but instead a random permutation of image values:

$$\log L_1 = E\left[ -\frac{1}{N_p} \sum_{p=1}^{N_p} \frac{\left(Z_{i,p} - \mathfrak{S}_{N_r}\left(Z_{j,p}\right)\right)^2}{\sigma_{Z_{i,p}}^2 + \sigma_{Z_{j,p}}^2} \right] \tag{4}$$

Here, $N_p$ is the number of *z-score* values in the field and $N_r$ is the number of iterations for which the random permutations, $\mathfrak{S}_{N_p}$, of the *z-score* values of image $j$, $Z_j$, is conducted. This $N_r$ was heuristically set to 100. In addition, $\sigma_{Z_{i,p}}^2$ and $\sigma_{Z_{j,p}}^2$ are the sensor and acquisition mode-specific variances per pixel $p$ in *z-score*-equivalent values, equal to pixel-specific $\frac{\varsigma_{\text{TOTAL},p}^2}{\sigma_i}$ and $\frac{\varsigma_{\text{TOTAL},p}^2}{\sigma_j}$, respectively. The corresponding total noise budgets were presented in Table 3. For LS8, noise is based on the RMSE value as reported in Section 2.3, without taking into account noise reduction from spatial averaging. The resulting consistency metric, with $-1 \leq C \leq 1$ defining the range, is equal to the upper limit (one) for two equal images, is equal to zero for two perfectly random images, and is equal to the lower limit for two inversely correlated images. The pixel variances give importance to the samples by weighing the differences between pixel values. Since pixels with high backscatter are generally associated with low variances (see Table 3), more weight is given to the outcome of the absolute differences between the pixels. Hence, comparisons between images with distinct and similar patterns result in a higher consistency value as compared to images with similar but less distinct patterns. All SAR images are labelled for rain events; for the optical images, the potential effects of rain conditions on consistency were not analyzed and hence the images are not labelled.

## 3.3. Intersensor Imagery-Consistency Analysis

From previously presented consistency matrices per sensor, we computed an average *z-score* image from the *z-score* images present in the time windows with highest consistencies. The similarities of these average *z-score* images between the SAR and optical sensors were measured by the (Pearson) correlation.

## 3.4. Intrafield Variability Analysis

In addition to the relationship of field-averaged remote sensing signals with sugarcane biomass (Section 3.1), we also analyze the relation between spatial patterns in the averaged *z-score* images and the intrafield sugarcane biomass estimates, quantified as TCH. This is examined by two approaches: based on the TCH estimates at the ESU locations (varying in space and time), and based on the TCH estimates at the intensive-measurement (IM) locations (varying only in space).

### 3.4.1. ESU locations

Since the ESUs cover approximately one LS8 resolution cell, the average *z-score* images are based on remote sensing images that were resampled to a 30 by 30 m grid. For S1, given its coarse resolution, this results in oversampling. For WV2, the original resolution *z-score* values were averaged that were located within the ESU dimensions. The selections of the remote sensing images are based on the time windows with maximum consistency, as were defined in Section 3.2. For each sensor, the *z-score* value was extracted that was nearest to the corresponding ESU locations. In addition, per sensor and per ESU, the range of TCH values was taken from the start until the end of the time window based on the modeled TCH graphs (see Figure 4). For WV2, both images were analyzed.

### 3.4.2. IM locations

Biomass estimations along the spatial profile in Field F2 (see the right part of Figure 4) were correlated to two sets of *z-score* values:

- Based on averaged *z-scores* within the window of maximum temporal consistencies.
- Based on the single image per sensor that was closest to the intensive-measurement acquisition time. For the WV2 image (acquired 103 days after start of growth, on 25 January 2015), the original-resolution *z-score* value was taken closest to the recorded GPS location.

## 4. Results

### 4.1. Backscatter Analysis

For C-band HV (and, similarly, for C-band HH, which is not visualized here) the saturation point of the signal occurred at approximately 25 tons/ha (90 cm stalk height) for rain-affected signals and approximately 35 tons/ha (120 cm stalk height) for signals not affected by rain (see Figure 10). These saturation points coincided with the canopy-development phase, after which the increase in leaf length stagnated and the grand growth phase of the stalk commenced, with increasing stalk biomass dominance over leaf biomass (see Figures 1 and 5). Afterward, both signals that were affected by rain and not affected by rain tended to converge. In addition, signals affected by rain showed backscatter values for bare soil and marginal canopy that were close to those for fully mature sugarcane. This is contrary to signals not affected by rain, which showed clear differences.

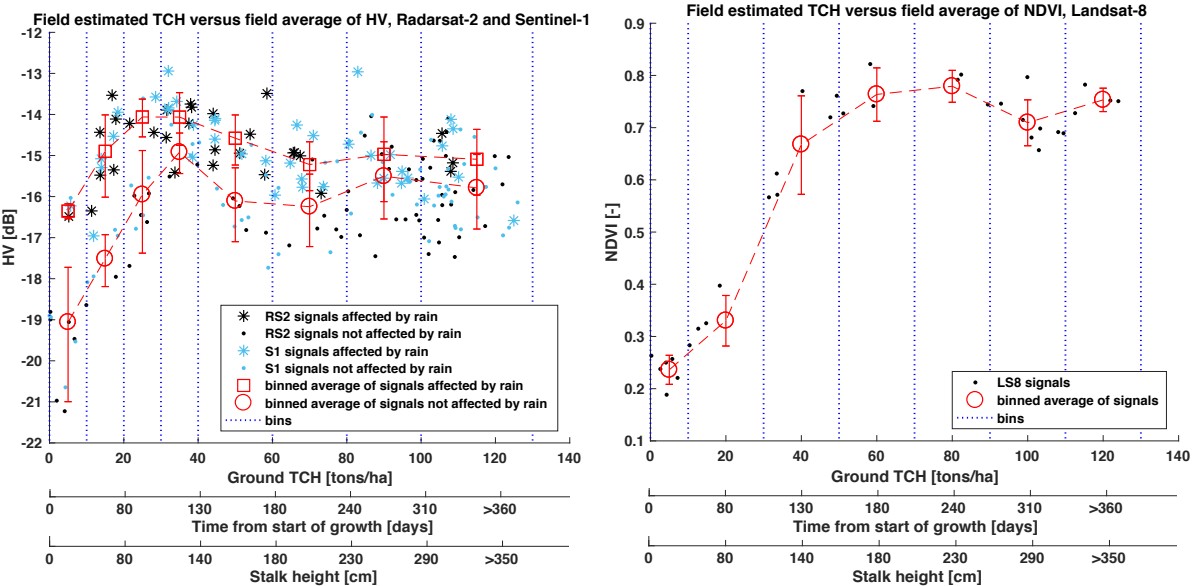

**Figure 10.** RS2 and S1 combined HV (**left**) and LS8 NDVI (**right**) signals versus binned values of estimated TCH, showing the effect of rain events on the signal. RS2 values were corrected for beam angles to the S1 angle. Additionally, the approximate and indicative axes of time from start of growth and stalk height are added. Bars illustrate standard deviations of remote sensing signals within each each TCH bin.

For both HV and HH, the median of the signals affected by rain after the saturation point significantly differed (with 95% confidence) from the median of the signals not affected by rain after the saturation point (see Figure 11). In addition, the figure demonstrates that the magnitude of this difference is comparable for both polarizations.

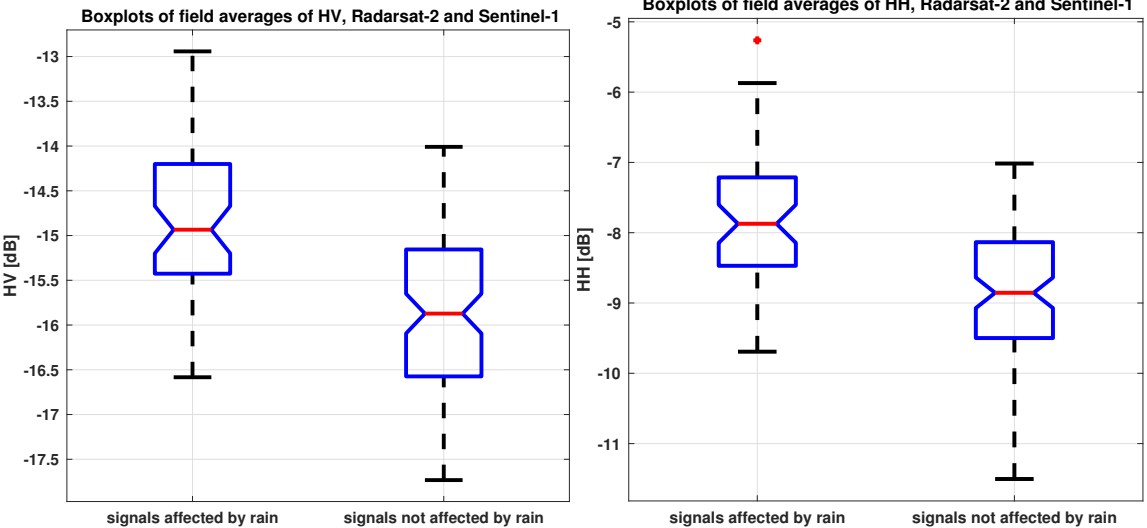

**Figure 11.** Box plots of RS2 and S1 HV (**left**) and HH (**right**) signals combined over sugarcane fields after their saturation points, showing the effect of rain events on the signal. Central (red) line indicates the median, and bottom and top box edges represent the 25th and 75th percentiles, respectively. Whiskers spanning to the most extreme data points were not considered outliers, and cover approximately 99% of the values. Notches represent the 95% confidence intervals of the median.

For NDVI, the saturation point occurred at 80 tons/ha (240 cm stalk height), which coincided with the transition from the grand growth phase to the maturing and ripening phase.

For ALOS-2 signals in HV and HH, signals did not show evident saturation points with increasing biomass (see Figure 12). Due to the limited number of available images, signals affected by rain and not affected by rain were combined for the averaging per bin. Nevertheless, similarly to C-band SAR, the L-band signals, especially in HH, showed sensitivity to surface wetness.

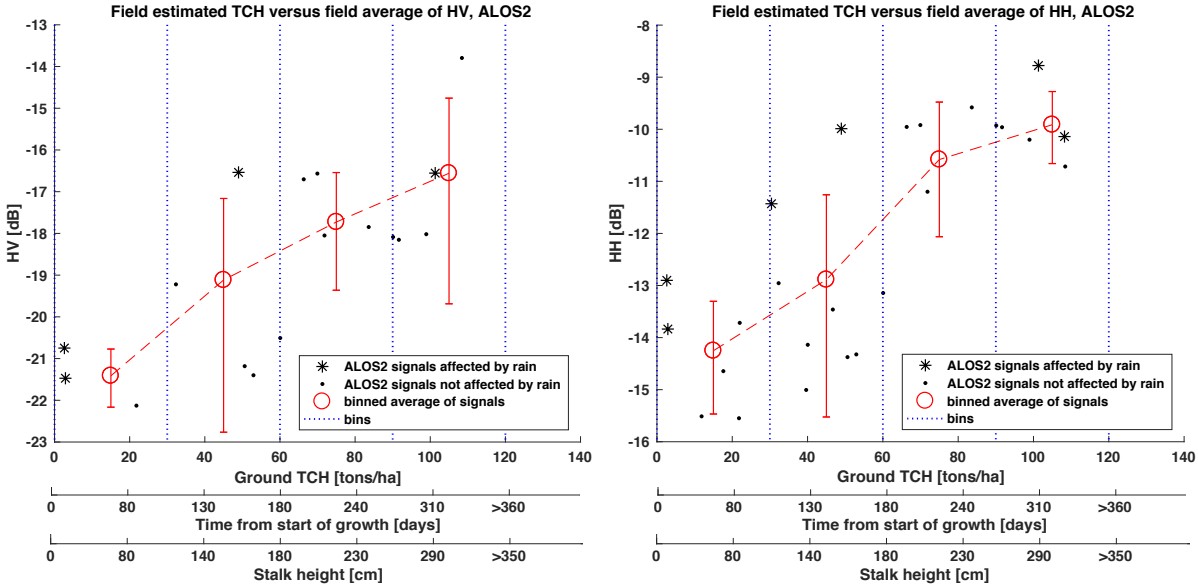

**Figure 12.** ALOS-2 HV (**left**) and HH (**right**) signals versus TCH, with signal averages taken from combined signals affected by rain and not affected by rain. Approximate and indicative axes of time from start of growth and stalk height were added.

## 4.2. Intrasensor Imagery-Consistency Analysis

From the consistency matrices between images over time (see Figure 13), different dynamics can be observed. First of all, the ranges of consistencies differ per sensor, whereby the consistency is highest for LS8, followed by ALOS-2, RS2 and finally S1. Also the time frame during which highest consistencies are observed differ.

For LS8 NDVI, images acquired during early crop growth showed relatively low consistency with later images. The image pair taken 63 and 95 days after growth was more similar due to the relatively small difference between the dates (two revisits' difference). The time window with the most similar images occurred from the late grand growth stage (approximately 210 days after growth) until harvest. The (not visualized) consistency of the WV2 NDVI images taken 103 and 301 days after growth (on 25 January and 11 August 2015, respectively) was 0.4, which is equal to the LS8 consistency between images taken on 95 and 303 days after growth, and lower than those between later LS8 NDVI images.

For ALOS-2, the highest consistency was found between two consecutively acquired images during the grand growth phase. The maximum value for ALOS-2 images was comparable to the highest consistencies found for LS8.

For the C-band SAR sensors, especially S1, most consistency values ranged between zero and 0.4, whereby the majority of the images showed a high degree of randomness in spatial patterns over time. For RS2, a window with higher values was found between approximately 200 and 330 days after growth. Before showing the details on this window and providing the corresponding explanation, we show a

more clear view on the variation in consistencies over time by taking the average over time windows for both sensors (see Figure 14). Each value in these matrices represents the average of the consistencies in Figure 13 that fall in the (triangular) window from the image indicated on the horizontal axis until the image indicated on the vertical axis. Through this representation, it could be found that images until approximately 200 days after start of growth were generally not similar with each other and hence showed a high degree of temporal randomness in spatial patterns. An exception is a minor time window that coincides with the period around the saturation point. For both C-band SAR sensors, the highest consistencies occurred for the images acquired during the second half of the grand growth phase until the maturing phase (i.e., between approximately 200 and 320 days after start of growth), after which the ripening phase commenced and images started to decorrelate. This time window commenced after the signal-saturation point (see Figure 10) when the backscatter decreased and coincided with the decrease in leaf biomass, though before severe leaf senescence occurred.

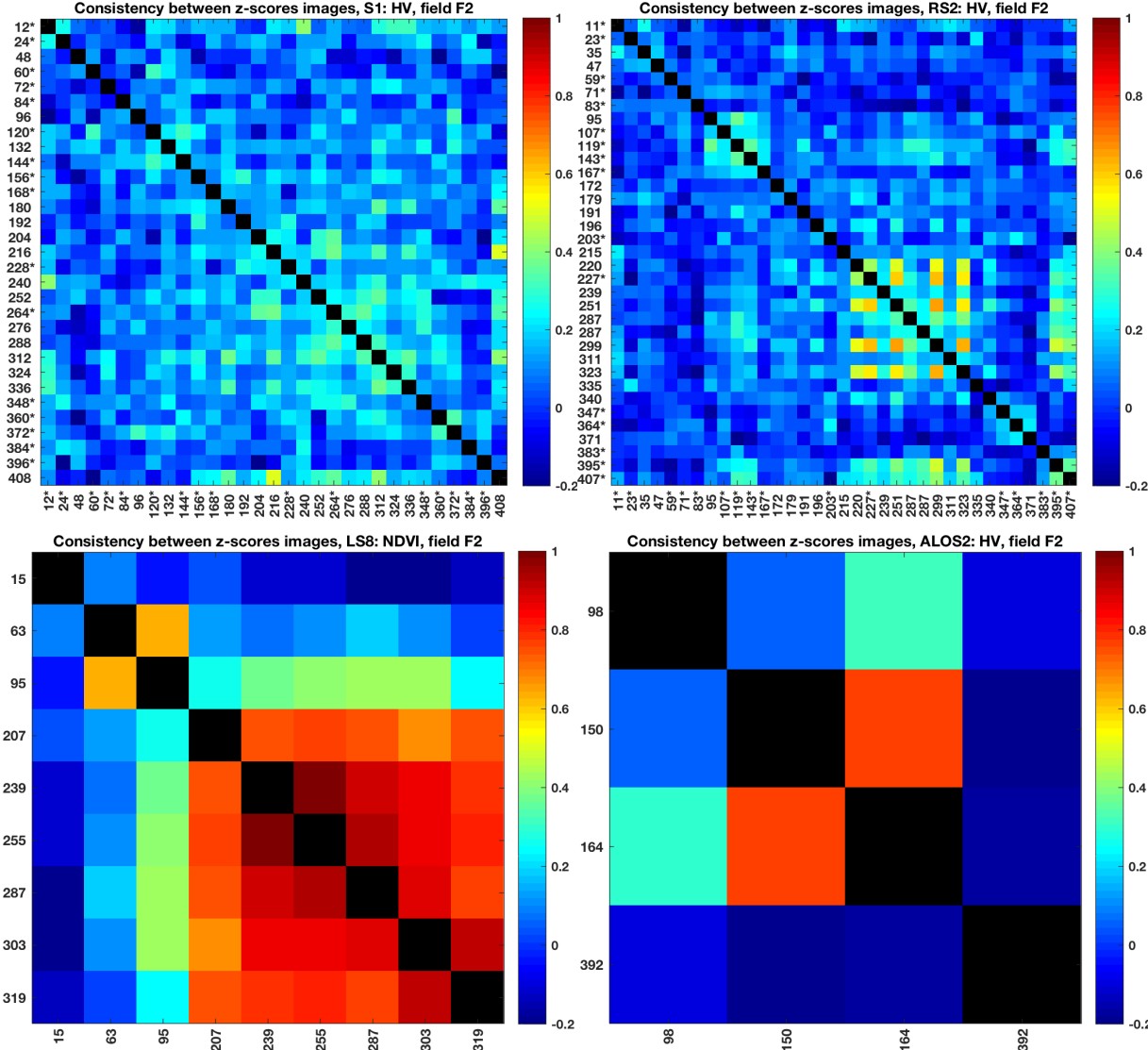

**Figure 13.** Consistencies of all image combinations for different sensors. Both axes show days from start of growth and asterisks indicate which acquisitions were rain-affected. Diagonal values were omitted and colored black. The harvest of this field was 419 days after start of growth.

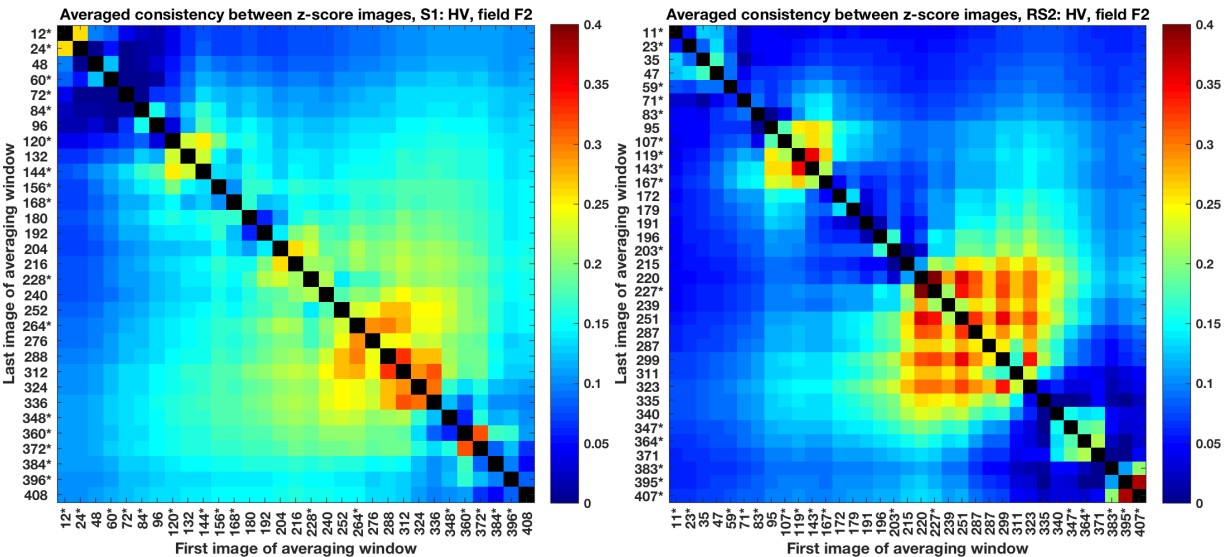

**Figure 14.** Consistencies for S1 HV (**left**) and RS2 HV (**right**) averaged over time windows based on the elements in Figure 13 with the first image on the horizontal axis and the last image on the vertical axis. Both axes present days from start of growth and asterisks indicate which acquisitions were rain-affected. Diagonal values were omitted and colored black.

The subset with highest consistencies for RS2 from Figure 13 accompanied by beam-mode information shows that the highest similarities were found between images taken in the FQ16W and S6 modes from ascending passes (see the left matrix of Figure 15). This is confirmed by the averaged consistencies for all beam-mode combinations within the window from 220 to 323 days after start of growth (see the right matrix of Figure 15). Contrary to the relatively high consistencies between these two modes, their consistencies with S7-ascending images are close to zero. This can be explained by the relative high incidence angle for S7, approximately 49°, with respect to the angles for FQ16W and S6, approximately 36° and 42°. The hour of acquisition for these modes was approximately similar (all around 21:00 UTC). The two images that were taken on the same day (287 days after growth) showed that changing beam modes (FQ21-descending and S7-ascending) results in low image similarity. In addition to their difference in incidence angle and look direction, low consistency may also be caused by the FQ21-ascending mode acquiring in the morning (when potential dew causes more surface wetness) versus the S7-ascending mode acquiring in the late evening (when the heat of the day causes lower moisture content).

Specifically regarding the effect of rain on similarities, the box plots of Figure 16 show that the median of the consistencies of the no-rain-to-no-rain (NR2NR) image pairs was higher than the median of the combined no-rain to rain (NR2R) and rain-to-no-rain (R2NR) image pairs, and the rain-to-rain (R2R) image pairs. This can be ascribed to the saturation effects and smaller spread of percentiles (see Figures 10 and 11) for the rain-affected images.

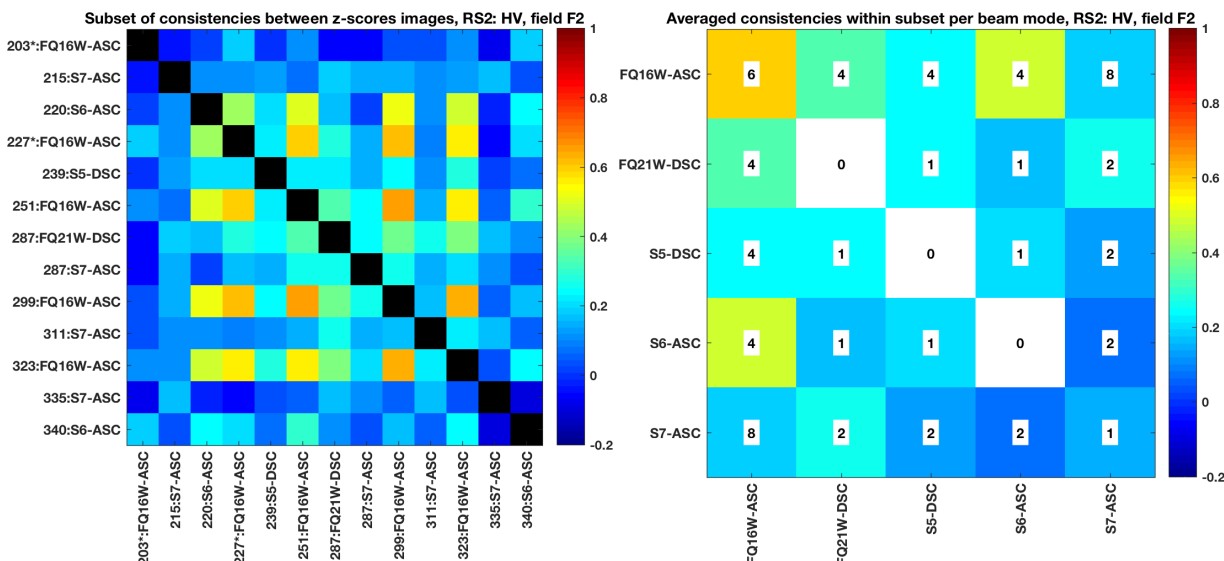

**Figure 15.** Subset of the RS2 HV consistency matrix of Figure 13 accompanied by beam-mode information (**left**). Both axes present days from start of growth and asterisks indicate which acquisitions were rain-affected. Diagonal values were omitted and colored black. In addition, averages of the consistencies between beams with the number of unique image pairs are displayed in each cell (**right**). White cells have no value as no image pair exists for those specific modes.

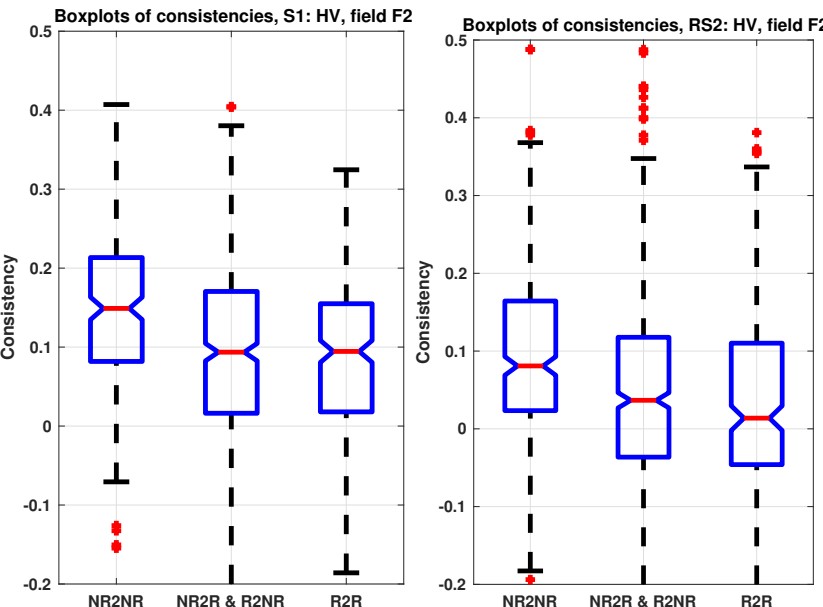

**Figure 16.** Box plots of all consistencies for S1 HV (**left**) and RS2 HV (**right**) per rain-condition pair, representing no rain to no rain (NR2NR), no rain to rain and rain to no rain combined (NR2R and R2NR), and rain to rain (R2R). Central (red) line indicates the median, and bottom and top box edges represent the 25th and 75th percentiles, respectively. Whiskers span to the most extreme data points not considered outliers and cover approximately 99% of the values. Notches represent the 95% confidence intervals of the median.

Hence, it may be deduced that the most similar images acquired by the C-band sensors could be found during the earlier-mentioned time window for images that were not affected by rain. In addition,

specifically for RS2 images that were acquired with changing beam modes, images with a high incidence angle and changing orbital passes (i.e., ascending and descending) should be avoided. Based on these findings, the following sensor-specific criteria may be summarized for selecting images that showed maximum temporal consistency:

- S1 HV: time window from 204 days until and including 336 days after start of growth, and only with nonrain-affected images, resulting in nine images.
- RS2 HV: time window from 220 days until and including 323 days after start of growth, only with nonrain-affected images and only for images acquired in the FQ16W and S6 modes, both ascending, resulting in four images.
- ALOS-2 HV: time window from 150 days until and including 164 days after start of growth, only with nonrain-affected images, resulting in two images.
- LS8 NDVI: time window from 207 days until and including 319 days after start of growth, regardless of rain conditions, resulting in six images.
- WV2 NDVI: only the image taken 301 days after start of growth, because Field F2 qas partly masked by clouds in the image taken 103 days after start of growth (see Figure 3).

These *z-score* images were averaged for each sensor and visualized in Figure 17.

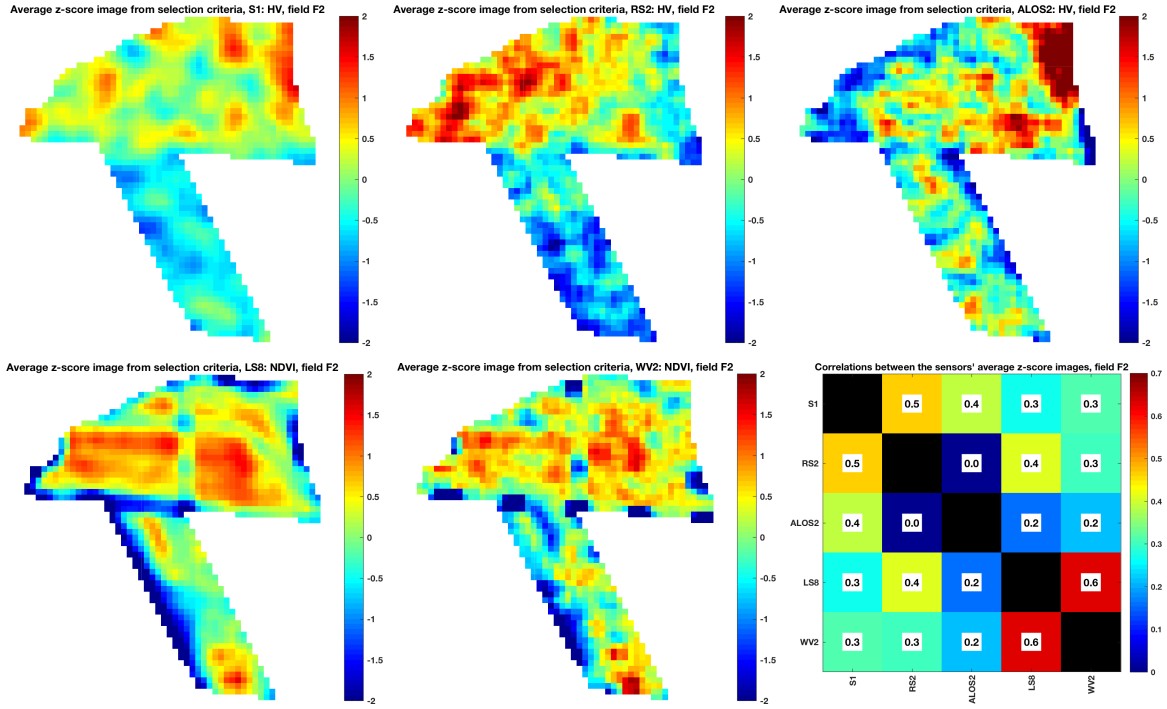

**Figure 17.** Field F2 average *z-scores* from images specified by the selection criteria for the different sensors (field plots), and corresponding correlations of these images between the different sensors (bottom right). Correlation coefficients are displayed in each cell, and diagonal values were omitted and colored black. The north direction is up, and the field spans approximately 1.4 km northward by 1.7 km eastward.

## 4.3. Intersensor Imagery-Consistency Analysis

The highest correlations between the averaged *z-score* images (see the lower right matrix in Figure 17) occurred between the optical sensors, followed by correlations between the C-band sensors. The images of both LS8 and WV2 showed agreement on the intrafield road features (see also the NDVI image of Field F2 in Figures 3 and 7) and regional differences between the northern and southern part of the field.

For the C-band SAR sensors, the images showed locally contrary behavior in the northern part of the field, i.e., the eastern part featured high values for S1 and low values for RS2, and conversely for the western part. More corresponding behavior was found in the southern part of the field. On a coarser level, the northern part of the field showed higher values than the southern part for both sensors.

For ALOS-2, the patterns with the highest *z-score* values occurred in the northeastern part of the field, coinciding with the lowest field elevations. As was described in Section 1, soil backscatter contribution, which is particularly dependent on soil-moisture content, is under standing vegetation conditions higher for L-band SAR than for C-band SAR. Consequently, higher values in the field's northeastern corner may be associated to high soil and plant moisture. The patterns in the southern part are not evidently consistent with image patterns of other sensors, and ground observations cannot give conclusive explanations for this.

### 4.4. Intrafield Variability Analysis

#### 4.4.1. ESU Locations

Table 4 shows that the differences in *z-score* values per sensor are not indicative of ESU biomass differences, except for the WV2 image taken 103 days after growth. Due to this sensor's fine resolution and hence higher geolocation accuracy, in combination with the absence of speckle noise, the signals more accurately present the state of vegetation at such small locations. The difference between the August acquisitions is that these two images were taken before and after the saturation point.

**Table 4.** TCH ranges between the first and last image in window of images of estimated TCH and associated *z-score* values per sensor for all ESUs of Field F2. [1] WV2 image taken 103 days after start of growth; [2] WV2 taken 301 days after start of growth.

| Name | ESU1 | ESU2 | ESU3 | ESU4 |
|---|---|---|---|---|
| TCH range S1 | 67 to 83 | 70 to 87 | 91 to 121 | 92 to 122 |
| *z-score* S1 | 0.1 | 0.3 | −0.5 | −0.2 |
| TCH range RS2 | 69 to 82 | 72 to 86 | 96 to 119 | 98 to 121 |
| *z-score* RS2 | 1.4 | −0.5 | 0.1 | -0.9 |
| TCH range ALOS-2 | 66 to 66 | 70 to 70 | 76 to 79 | 76 to 80 |
| *z-score* ALOS-2 | 0.5 | −0.4 | −0.1 | −0.6 |
| TCH range LS8 | 68 to 82 | 70 to 86 | 92 to 119 | 93 to 120 |
| *z-score* LS8 | −0.2 | 0.1 | 0.3 | −0.9 |
| TCH range WV2 [1] | 80 | 83 | 116 | 118 |
| *z-score* WV2 [1] | 1.1 | 1.1 | 1.2 | 1.2 |
| TCH range WV2 [2] | 80 | 83 | 116 | 118 |
| *z-score* WV2 [2] | 1.1 | 1.1 | 1.0 | 1.1 |

#### 4.4.2. IM Locations

Based on the correlations (see Table 5), it may be deduced that none of the time window-averaged *z-score* images was indicative for locating biomass differences along such a measurement profile. This can be expected due to the low correlations between images close to the IM date, as shown in Figures 13 and 14, as well as the high local variability within a 30 by 30 m resolution cell and geolocation accuracy. Regarding the *z-score* images that were closest to the measurements, the first WV2 image (acquired 103 days after start of growth) showed the highest potential for indicating these differences in the field.

**Table 5.** Correlation coefficients between estimated TCH values at the intensive measurement locations and *z-scores* of images per sensor.

| Sensor | Window Averaged | Closest Date |
|--------|-----------------|--------------|
| S1 | 0.1 | 0 |
| RS2 | 0 | −0.1 |
| ALOS-2 | 0.2 | 0.3 |
| LS8 | −0.1 | 0 |
| WV | −0.2 | 0.6 |

## 5. Discussion

The results presented the effects of sugarcane biomass growth, precipitation, and sensor characteristics on remote sensing signals, in particular from C-band SAR, L-band SAR, and optical sensors. These effects were first examined through the relationship between field-averaged remote sensing signals and sugarcane biomass. Second, remote sensing imagery was analyzed for the statistical coherence of spatial features in time. The identification of time windows per sensor allowed for the effective extraction of the most consistent patterns. It was found that rain events have their effect on this as well, specifically for the C-band SAR sensors. Subsequently, the patterns were extracted and compared between the various sensors. Finally, the relationship between these patterns and measured intrafield sugarcane biomass was investigated. Hereafter, these elements will be compared and discussed from an integrated perspective.

### 5.1. Backscatter Analysis

The first variant of sugarcane productivity monitoring was presented through the relationship between field-averaged remote sensing signals and sugarcane biomass. The saturation points of the signals differ per wavelength and, for SAR, are dependent on the wetness conditions of the surface. For C-band signals, saturation occurs earlier when affected by rain conditions than when unaffected by rain conditions. After saturation, backscatter values decrease, which can be ascribed to a combination of two effects. First, the increasing dominance of stalk biomass over leaf biomass (see Figures 1 and 5) causes a reduction in the depolarizing effect of volumetric scattering and an increase in the vertical copolarizing effect due to higher microwave interaction with the vertical stalks, and thus an attenuation of the cross-polarized signals. Second, there is a decreasing contribution of microwave interaction with water volume in the plant due to a decline in wet stalk content and leaf senescence (see Molijn et al. [24]). Both interactions were also (partly) reported by Baghdadi et al. [6,21] and further supported by the biophysical developments observed in Simoes et al., and Vieira et al. [5,17,19]. Even though the signals after the saturation points decreased and the two signal profiles at high biomass values tended to converge, the profile affected by rain clearly differed from the profile unaffected by rain. Furthermore, the perceived converge of the two profiles with increasing biomass values suggests that the contribution from microwave interaction with the volumetric features dominates over the contribution from surface wetness. Hence, the response of C-band SAR signals to surface wetness is a function of vegetation biomass. Additionally, the results showed comparable C-band backscatter values for bare soil when affected by rain with respect to the values for mature cane irrespective of rain conditions. Such confusion between signals acquired under wet-soil conditions and under mature sugarcane conditions was also observed for X-band SAR, for example by Baghdadi et al. [21]. Furthermore, Molijn et al. [54] showed that C-band backscatter fluctuations in time, caused by precipitation events, could be used for improving the classification of vegetation conditions. Inversely, when vegetation conditions are known, these signals may be used for detecting rain events and, in combination with the relationship between temporal backscatter development and biomass water content, may enhance sugarcane drought monitoring.

NDVI signals saturate at a later stage (approximately at 80 tons/ha, 240 cm stalk height), and also afterward show a signal decrease. This coincides with the decrease in leaf biomass (see Figure 5) and ongoing senescence. ALOS-2 cross-polarized and copolarized signals did not reveal clear saturation points and showed longer sensitivity to sugarcane growth, which can be ascribed to the longer wavelength [6]. Additionally, the copolarized signals (HH) suggest there was also an effect of rain on backscatter intensities.

## 5.2. Intrasensor Imagery-Consistency Analysis

The time windows during which image features were most consistent differ between the various types of sensors. In addition, image consistencies were found to be generally lower for SAR sensors than for the optical sensors, with lowest values for S1 imagery and a relatively high value for ALOS-2. For the C-band SAR sensors, temporal averaging of the consistency metric allowed for the identification of these time windows. These were, as may be expected, comparable between S1 and RS2 imagery. A minor window that was identified coincided with the period around the saturation point. The general and principal window spans from the late grand growth phase until the maturing phase, after which the ripening phase commenced and images started to decorrelate. As was discussed in the previous section, during this time frame the C-band SAR signals showed, on average, lower backscatter values than at saturation point. For both sensors, images that were not affected by rain showed higher consistencies than when affected by rain. Consequently, it may be deduced that rain induces fluctuations in signal backscatter causing increased disorder in spatial patterns. For RS2 specifically, image pairs that were acquired under different beam modes should be avoided. Furthermore, one should be aware of the time of acquisition, as it may increase the chance of surface wetness (e.g., caused by morning dew).

For ALOS-2, it was found that the spatial patterns from two images taken consecutively within one revisit, not affected by rain and during the grand growth phase, showed the highest temporal agreement, comparable to values that were found for the optical sensors. Regarding LS8, the most effective time window spanned from the late grand growth phase until harvest. Finally, consistency between the two WV2 images was similar to what was observed for the LS8 images acquired during the same growth phase.

From a combined view on the time windows for the C-band and optical sensors, it may be noticed that the highest consistencies generally occurred during the final growth stages. Based on the descriptive and mathematical definition of the term consistency given in Section 3.2, specifically on the effect of pattern distinctiveness, the higher values were likely caused by increased spatial-biomass differences over the course of sugarcane growth, which is supported by the measurements in Figure 4 and by the observed distinct patterns in the averaged *z-score* images (see Figure 17).

## 5.3. Intersensor Imagery-Consistency Analysis

The comparisons between the most consistent patterns extracted for each sensor showed that highest agreements are found between the optical sensors. Local intrafield roads and the regional patterns in the field are especially common. For C-band SAR, agreements were generally observed for regional patterns. Spatial agreements and differences between the optical and SAR images should especially be further investigated. Special attention should be paid to the suitability of comparing the averaged *z-score* patterns when time windows differ between the sensors.

## 5.4. Intrafield Variability Analysis

Despite the limitations of the ground measurements, the results indicated that high-resolution optical-satellite imagery has the most potential for mapping intrafield sugarcane productivity differences.

From an integrated perspective on previous findings, it is recommended to focus on the relationship between the physical properties of the signals and the extracted patterns. Since patterns from optical

imagery may be linked to canopy conditions, and patterns from SAR imagery may be linked to volume and soil-moisture conditions, their integration may indicate fertility and moisture issues. The different sensitivities may also be used for the detection of field anomalies, including gaps and plant substitution (such as weed infestations). This should be supported by more detailed ground measurements, specifically on soil moisture, plant moisture, and sugarcane biomass. It is recommended to focus on taking these measurements during the overlap between the specified time windows, from the late grand growth phase until the early maturing phase.

## 6. Conclusions

In conclusion, for sugarcane productivity mapping, satellite imagery offers several opportunities. For monitoring field-averaged sugarcane-biomass growth, signals from both SAR and optical sensors can be used effectively during certain growth stages. C-band SAR imagery generally offers smallest time frames for growth monitoring, especially when the signals are affected by precipitation. Adversely, for L-band SAR imagery, saturation effects were not observed; hence, it offers growth-monitoring potential during the entire growth period. Nevertheless, L-band signals also appeared to have been affected by precipitation events. It is recommended that the precipitation sensitivity of both cross-polarized and copolarized signals from C-band and L-band sensors should be further explored for the detection of these events and therefore for drought monitoring as well. Optical imagery offers effective time frames that range between those from C-band and L-band imagery.

For monitoring intrafield sugarcane-biomass differences, the presented analyses gave an indication of the selection of images to investigate. Local and regional patterns can be obtained through temporally averaging spatial features within these windows. Especially for the SAR imagery, this reduces the noisy appearances of features and facilitates analysis for physical interpretation. Outside of these time windows, high variations of spatial-feature locations in time complicate this assessment. Comparing the resulting patterns between sensors shows that the highest agreements were found between the C-band SAR sensors and between the optical sensors. Furthermore, the intersensor comparison revealed that, in general, pattern agreements are found on a regional scale and differences on a local scale. As such, integration between sensors can be used for obtaining additional indications of regional productivity differences that may be related to water and fertility concerns. The mapping of local differences may serve as indications for local variations in field quality that may be related to plant failure and substitution (such as weed infestations). The collected ground measurements were too limited for proper validation of these specific potentials. Nevertheless, the distinctive interaction mechanisms between remote sensing signals and observed features may be exploited. For example, the sensitivity of SAR signals to water content and volumetric plant properties could be used for the detection of plant substitution where optical signals could be less successful due to the limited sensitivity to plant geometry. Several limitations associated to satellite imagery should be respected during these analyses, in particular, differences in time windows between the sensors, signal noise (especially for SAR imagery), and spatial resolution.

Finally, through relating patterns to the collected ground measurements, it was found that high-resolution optical images are most effective for providing indications on intrafield biomass variations. When similar ground campaigns are henceforth undertaken, it is recommended to acquire spatially dense plant and moisture measurements, including plant failure and plant substitution, on the scale of the SAR sensors' spatial resolutions from the late grand growth phase until the early maturing phase.

**Author Contributions:** R.A.M. conceived of and designed the study, designed and carried out the ground-measurement campaign, designed the methodologies, analyzed and interpreted the data and results, and wrote the manuscript. L.I. assisted in designing and carrying out the ground-measurement campaign, framing the statistical analyses, interpreting the data and results, reformulated parts of the manuscript, and reviewed the manuscript. J.V.R. established a relation and an agreement with the owner of the sugarcane fields (together with Rubens Lamparelli), assisted in validating and approving the results, and reviewed the manuscript. R.F.H. initiated the overall project and funding, endorsed the design, and approved the manuscript.

**Funding:** This research received no external funding.

**Acknowledgments:** The authors would like to acknowledge the European Space Agency (ESA) for providing the Radarsat-2 data under the framework of project C1P.16849, the Google Earth Engine team for providing access to the Sentinel-1 and Landsat-8 data, the Japan Aerospace Exploration Agency (JAXA) for providing the ALOS-2 data under Research Announcement-6, number 149250, and the Brazilian Space Agency (INPE) for providing the Canasat maps of the São Paulo region. In addition, we would like to thank the colleagues at FEAGRI, Unicamp, for their assistance with taking the ground measurements, in particular, Diego della Justina, Carlos Wachholz de Souza, Walter Rossi Cervi, Rubens Lamparelli, and Ali Mousivand. Finally, we acknowledge and are grateful for the access to and safety in the sugarcane fields provided by Jefferson Rodrigo Batista de Mello, Eduaro Caetano Ceará, Pedro Lian Barbieri, and Alex Thiele Paulino. The work was carried out in BE-Basic project FES0905, and partly carried out within the framework of the joint BE-Basic FAPESP project 2013/50942-2.

**Conflicts of Interest:** The authors declare no conflict of interest.

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
