# Peer review of "Sugarcane Productivity Mapping through C-Band and L-Band SAR and Optical Satellite Imagery"

_remotesensing, doi:10.3390/rs11091109_

Round 1
Reviewer 1 Report
The paper analyzes the behavior of SAR (C band and L band) and optical data with growing sugarcane and the influence of precipitation events. The temporal consistency is also studied. Even if the paper is well written in general, some points should be improved to allow the reader a better understanding of the paper content. In addition, table (s) with values of in situ measurements is necessary.
Major comments:
- Figure 10: It is not clear how the points affected by the rain (or not affected) are extracted. Please give the definition of "points affected or not affected by rain" ==> This correspond to rain on the acquisition day or vene a few day before the SAR acquisition and from what quantity of rain?
- It is necessary to add a table that summarizes the in situ measurements. For example Biomass [min, mean, max] ...
Its is necessary to clarify a little more what it means to say the term "consistency"
- Line 418 "L-band ....standing vegetation". For me this sentence is not correct. L-band is not more sensitive but the part of the soil in the total signal is higher in L-band than in C-band.
- The conslusions of the paper are not clearly established. It is necessary to be more clear about the main conclusions of the study by trying to use a little less the term consistency.
Minor comments:
- Line 8: the meaning of "differences" is not clear.
- Need to add in the introduction some references using SAR in X band.
- Figure 3, Field F1: the form of the field F1 is not the same that the form in Figure 2. In addition, the limit of the field is not clear in Figure 3.
- Figure 5, right plot: If your Biomass = TCH please the same terminoloy. In addition the unit is not kg but kg/ha. It could be better if you can use the same unit in Figure 4 and in Figure 5 for the biomass (tons/ha or kg/ha).
- Table 2: remove "Revisit". Acquisition time is suffisant. Also, remove the column "Angle product".
- Line 152: remove "also denoted as ALOS2"
- Line 178, normalization of the radar signal: please give a reference and explain why you use cos(teta) and no cos²(teta) or cos^a(teta).
- line 230: since a long time the SRTM DEM es available with a spatial resolution of 30 m
- Line 315: for me 35 tons/ha and no 40 tons/ha
- Figure 11: you explain several times that HV is better than HH but figure 11 shows similar performance with HH and HV
Author Response
Thank you for the review. Please find my responses to the comments in the attached document.

Reviewer 2 Report
Dear Authors
I revised the manuscript “Sugarcane productivity mapping through C-band and L-band SAR and optical satellite imagery” submitted to the Remote Sensing Journal. The paper is interesting and evaluated five SAR and optical sensors to map sugarcane productivity in Sao Paulo, Brazil. However, I have some minor concerns, which need to be conferred before final publication.
Broad comments:
· I would suggest to include results and a final conclusion sentence in the Abstract.
· Please increase the size and resolution of images in whole manuscript.
· Please separate Results and Discussion chapter and compare your results with previous findings elaborately in the Discussion chapter.
· Please add the Summary with Discussion chapter and separate Conclusions.
Specific comments:
· Line 30, please include a space before “Review works by …..”. In the same line, please use the citation as “McNairn and Brisco, and Susan et al. [24, 44] and a study …..”. Same for whole manuscript.
· Line 96, please delete one “.” in the sentence “They are located ……”.
· Line 98, please rewrite as “The agriculture in ….”
· Table 2, revisit time of LS8 is 16 days, not 17 days.
· Line 151, please use (LS8) after Landsat-8.
· Line 152, please use (RS2) after Radarsat-2. Same for the whole manuscript.
· Line 161, please use (WV-2) after WorldView-2. Same for WorldView-3.
Author Response

(The authors gave the same response as above.)

Reviewer 3 Report
Thank you for the work and paper. Please find my comments in attached manuscript.

Author Response

(The authors gave the same response as above.)

Reviewer 4 Report
This manuscript examine the effects of sugarcane biomass growth, precipitation and sensor traits on remote sensing signals from three SAR and two optical sensors for sugarcane fields in Sao Paulo state, Brazil. The effects were shown by establishing a relationship between field-averaged remote sensing signals and sugarcane biomass. Further, the temporal consistencies between satellite images over several sugarcane fields have been analyzed. Finally, a list of conditions during which the acquisition of satellite imagery is most effective for monitoring the sugarcane productivity differences has been provided.
The paper is well written and logically explained, however, few comments needs to be incorporated;
1. Figure 13 and 14 requires the x and y axis labels to be shortened, better to represent by the short keys.
2. The conclusion needs to be re-written clearly and precisely.
Author Response

(The authors gave the same response as above.)

Round 2
Reviewer 1 Report
Authors considered my comments.
Remain the comment in lines 428-430. The sentence "As described in section1, ... , ... standing vegetation" needs to be rewritten more correctly.
Author Response
Thank you for the latest review. Please find my responses to the comments in the attached document.

Reviewer 3 Report
Thank you for the revised version. I have few minor comments. Please find the comments in the manuscript attached.

Author Response

(The authors gave the same response as above.)
